# Anti-Inflammatory Effects of Honeysuckle Leaf Against Lipopolysaccharide-Induced Neuroinflammation on BV2 Microglia

**DOI:** 10.3390/nu16223954

**Published:** 2024-11-19

**Authors:** Bitna Kweon, Jinyoung Oh, Yebin Lim, Gyeongran Noh, Jihyun Yu, Donggu Kim, Mikyung Jang, Donguk Kim, Gisang Bae

**Affiliations:** 1Department of Pharmacology, School of Korean Medicine, Wonkwang University, 460 Iksan-daero, Iksan 54538, Republic of Korea; 2Hanbang Cardio-Renal Syndrome Research Center, Wonkwang University, 460 Iksan-daero, Iksan 54538, Republic of Korea; 3Department of Herbology, College of Korean Medicine, Dong-Eui University, 176 Eomgwang-ro, Busan 47340, Republic of Korea; 4Research Center of Traditional Korean Medicine, Wonkwang University, 460 Iksan-daero, Iksan 54538, Republic of Korea

**Keywords:** *Lonicera japonica* Thunb., leaf of honeysuckle (LH), honeysuckle, neuroinflammation, microglia

## Abstract

Background/Objectives: Neurodegenerative disorders have emerged as a major global public health concern, and the burden is predicted to increase over time. Modulating neuroinflammation and microglial activity is considered a promising target for improving neurodegenerative disorders. The leaf of honeysuckle (LH), which has anti-inflammatory properties, has long been collected, regardless of the season, and used for medicinal purposes. However, research on its effects on neuroinflammation is scarce. In this study, we determined the neuroprotective effects of LH water extract by inhibiting microglial activation induced by lipopolysaccharide (LPS). Methods: The production or secretion of pro-inflammatory mediators was examined in LPS-exposed BV2 cells to ascertain the efficacy of LH water extract in improving neuroinflammation. In addition, the phosphorylation of mitogen-activated protein kinases (MAPKs) and the degradation of inhibitory κBα (IκBα) were analyzed to elucidate the regulatory mechanisms of LH water extract. Ultra-performance liquid chromatography (UPLC) analysis was conducted to identify the active component of the LH. Results: LH water extract suppressed the formation of inducible nitric oxide synthase (iNOS), nitric oxide (NO), and pro-inflammatory cytokines, including interleukin (IL)-1β and tumor necrosis factor (TNF)-α, in LPS-activated BV2 cells. LH impeded the activation of c-Jun N-terminal kinase (JNK). Moreover, chlorogenic acid was found in LH. Conclusions: The above findings suggest that LH water extract could improve neuroinflammation.

## 1. Introduction

With the increasing mortality and disability associated with global aging, neurodegenerative disorders exert a significant impact on global public health [1,2]. In addition, the socioeconomic impact of neurological disorders is gradually increasing, causing a high economic burden, with expenditures for major neurodegenerative disorders reaching 800 billion dollars in the United States alone [3]. As the elderly population in the United States is expected to exceed 80 million by 2050, providing appropriate treatment and prevention strategies for neurodegenerative disorders are emerging as important issues [3]. Neurodegeneration is a multifactorial disease in which aging-related genetic, environmental, and endogenous factors are considered major basic mechanisms; however, more needs to be identified to solve the problem [4]. In general, the pathophysiological mechanisms of neurodegenerative disorders include abnormal protein folding, oxidative stress, mitochondrial dysfunction, and neuroimmune processes [5]. Although neuroinflammation initially protects the brain, excessive or persistent inflammatory responses can interfere with regeneration, leading to neurodegenerative disorders [6].

Honeysuckle (*Lonicera japonica* Thunb.) has been traditionally used as a medicinal herb in the Korean medicine, and has been named differently, depending on its medicinal part. In the Korean Pharmacopoeia (KP), the flower buds or flowers that have begun to bloom from the Caprifoliaceae honeysuckle are called Lonicerae Flos, and the leaves and vine-like stems are termed Lonicerae Folium et Caulis. The leaves and vine-like stems of honeysuckle (LSH) have little odor and a slightly astringent taste with a bitter aftertaste. Traditionally, they function to remove heat, detoxify, and disperse wind-heat. For this reason, they have been primarily used to treat arthritis caused by wind-dampness [7] (pp. 252–253). LSH contain diverse components, including phenolic acids, flavonoids, and iridoid glycosides. In particular, the chemical composition of the leaf of honeysuckle (LH) is considerably similar to that of the flower bud of honeysuckle (FH) [8,9]. According to a study [9], among the components contained in honeysuckle, chlorogenic acid was reported to be commonly and abundantly present in LH and FH.

A recent study reported that both LSH and FH have anti-inflammatory properties; however, LH results in better improvement than other parts [10]. In addition, chlorogenic acid is known to have excellent anti-inflammatory effects [11]. To date, research objectively demonstrating the efficacy of each leaf, stem, and flower bud of honeysuckle is insufficient. Moreover, in the market, the flower buds, stems, and leaves of honeysuckle are distributed without being properly distinguished [12]. The above-mentioned chemical similarity and pharmacological effects indicate that LH could be used as a substitute for FH, or it could be even more effective than FH. However, relatively little pharmacological research on LH has been conducted.

Several studies suggest the potential of FH to improve neuroinflammation [13,14]. However, the efficacy of LH water extract in ameliorating neuroinflammation has not yet been evaluated. Hence, this study focuses on assess of the neuroinflammation-improving effects of LH water extract. In addition, the regulation of inflammatory mediators and mitogen-activated protein kinases (MAPKs) signaling was evaluated using lipopolysaccharide (LPS)-activated BV2 microglial cells to study the inhibitory effect and its regulatory pathways of neuroinflammation.

## 2. Materials and Methods

### 2.1. Preparation of LH Water Extract

LH was purchased from the KOREA HERB TEA & FOOD LABORATORY (Jeju, Republic of Korea). A reflux decoction method was adopted to obtain the water extract of LH. In brief, the dried LH was ground and pulverized. Distilled water (600 mL) was added to 30 g of LH powder, and the reflux extraction was performed in boiling water at 100 °C for 2 h. The extract was filtered with Whatman filter paper (No. 2), freeze-dried, and stored in a freezer at −20 °C before use. The final extraction yield was 24.13%.

### 2.2. Cell Culture

BV2 microglial cells were acquired from AcceGen (Fairfield, NJ, USA). The RPMI medium 1640 (cat. no. 11875-093; Gibco, Thermo Fisher Scientific, Inc. Waltham, MA, USA) supplemented with 10% fetal bovine serum (FBS; cat. no. 16000-044, Thermo Fisher Scientific, Inc.) and 1% penicillin/streptomycin (cat. no. 15140-122, Thermo Fisher Scientific, Inc.) was used as the cell culture medium. The cells were cultured at 37 °C with 5% CO_2_.

### 2.3. The 3-(4,5-Dimethylthiazol-2-yl)-2,5 Diphenyl Tetrazolium Bromide (MTT) Assay

BV2 microglial cells between passages 3 to 5 were dispensed into a 24-well plate at a density of 2 × 10^5^ cells/well. The cells were cultured for 24 h with 0.01, 0.05, 0.1, 0.3, and 0.5 mg/mL of LH water extract. Thereafter, the MTT reagent was added, and the plate was incubated for 30 min. After removing the supernatant from the well plate, the formed formazan was dissolved in dimethyl sulfoxide (DMSO). Cell viability was assessed at an absorbance of 540 nm using SpectraMax M2 (Molecular Devices; San Jose, CA, USA).

### 2.4. Nitric Oxide (NO) Assay

Griess assay was conducted to assess nitric oxide (NO) concentration in the cell culture medium. BV2 microglial cells were seeded at 2 × 10^5^ cells/well and pre-treated with LH water extract at concentrations of 0.01, 0.05, 0.1, and 0.3 mg/mL. After 1 h, cells were co-cultured with LPS (1 μg/mL) at 37 °C and 5% CO_2_ for 24 h. Next, the supernatant was taken and reacted with Griess reagent at a 1:1 ratio. The concentration of NO was measured at 540 nm, and the amount of NO produced in the cells was assessed by calculation based on standard diluted sodium nitrite (NaNO_2_) at concentrations of 3.125, 6.25, 12.5, 25, 50, and 100 μM.

### 2.5. Real-Time Reverse Transcription-Polymerase Chain Reaction (RT-PCR)

BV2 microglial cells were placed in a 6-well plate at a density of 1 × 10^6^ cells/well and treated with LH water extract at concentrations of 0.01, 0.05, 0.1, and 0.3 mg/mL. After 1 h, cells were incubated with LPS (1 μg/mL) for 6 h. Total RNA was isolated from the cells using an Easy-Blue™ RNA extraction kit (cat. no. 17061; iNtRON Biotechnology, Inc., Sungnam, Republic of Korea). Gene Quant Pro RNA calculator (Biochrom Ltd., Cambridge, UK) was used for RNA quantification. Next, cDNA was synthesized using ReverTra AceTM qPCR RT Kit (TOYOBO; Osaka, Japan). DEPC-treated distilled water was used to dilute the synthesized cDNA at a 9:1 ratio. It was amplified using the SYBR Premix kit (Applied Biosystems; Waltham, MA, USA) with the following conditions: primary denaturation at 95 °C for 10 min, and then 40 cycles of denaturation at 95 °C for 15 s, and annealing at 60 °C for 1 min. All primers used were obtained from Cosmogenetech (Seoul, Republic of Korea), and the primer sequences were as follows: inducible nitric oxide synthase (iNOS): forward (5′-GTT GAA GAC TGA GAC TCT GG-3′) and reverse (5′-GAC TAG GCT ACT CCG TGG A-3′); cyclooxygenase (COX)-2: forward (5′-GGT GGC TGT TTT GGT AGG CTG-3′) and reverse (5′-GGG TTG CTG GGG GAA GAA ATG-3′); interleukin (IL)-1β: forward (5′-CCT CGT GCT GTC GGA CCC AT-3′) and reverse (5′-CAG GCT TGT GCT CTG CTT GTG A-3′); IL-6: forward (5′-CCG GAG AGG AGA CTT CAC AG-3′) and reverse (5′-CAG AAT TGC CAT TGC ACA AC-3′); tumor necrosis factor (TNF)-α: forward (5′-AAC TAG TGG TGC CAG CCG AT-3′) and reverse (5′-CTT CAC AGA GCA ATG ACT CC-3′); and glyceraldehyde 3-phosphate dehydrogenase (GAPDH): forward (5′-TGT GTC CGT CGT GGA TCT GA-3′) and reverse (5′-TTG CTG TTG AAG TCG CAG GAG-3′). GAPDH was used for normalization. The 2^−ΔΔCt^ method was used to analyze the relative gene expression [15].

### 2.6. Western Blotting

BV2 microglial cells were seeded in 6 cm dishes at a density of 5 × 10^6^ cells/dish, and half of them were pre-treated with 0.3 mg/mL of LH water extract for 1 h. With or without LH water extract pre-treatment, the cells were exposed to LPS (1 μg/mL) for 15, 30, and 60 min. Subsequently, the cells were lysed with the RIPA lysis buffer (iNtRON biotechnology, Sungnam, Republic of Korea) containing a 1% phosphatase inhibitor cocktail (cat. no. p5726; MilliporeSigma, Burlington, MA, USA) and 1% EZ block protease inhibitor cocktail (cat. no. K272-1; BioVision, Inc., Milpitas, CA, USA). Afterward, 40 μg of protein was loaded onto a 10% sodium dodecyl sulfate (SDS)-polyacrylamide gel. The membrane was incubated with primary antibody overnight at 4 °C. of the following antibodies were used: phospho-extracellular signal-regulated kinase 1/2 (pERK1/2) (#9101); ERK1/2 (#9102); phospho-c-Jun N-terminal kinase (pJNK) (#9251); JNK (#9252); phospho-P38 mitogen activated protein kinase (pP38) (#9211); P38 (#9212); inhibitory κBα (IκBα) (#9242); GAPDH (#2118). All primary antibodies were acquired from Cell Signaling Technology (Danvers, MA, USA). ERK1/2, JNK, P38, and GAPDH were used as loading controls for the expression of each protein. Goat anti-rabbit IgG (H + L)-horseradish peroxidase (HRP) (cat. no. SA002-500; GenDEPOT, LLC. Baker, TX, USA) was used as the secondary antibody. The protein expression was visualized using an enhanced chemiluminescence detection system (Amersham; Buckinghamshire, UK). ImageJ version 1.53 k (National Institutes of Health, Bethesda, MD, USA) was used to analyze the densitometry of western blots.

### 2.7. Ultra-Performance Liquid Chromatography (UPLC) Analysis

Standardized chlorogenic acid (cat. no. C3878) was acquired from Sigma-Aldrich (St. Louis, MO, USA). Methanol was used to dissolve the extract and prepare the standard. Ultra-performance liquid chromatography (UPLC) was conducted on an ACQUITY UPLC system (Waters, Milford, MA, USA) with an ACQUITY binary solvent manager pump and an ACQUITY PDA detector. The extract and standard were investigated under the following conditions: column was ACQUITY UPLC BEH Shield RP18 (2.1 × 100 mm, 1.7 μm; Waters, MA, USA) and the mobile phase consisted of distilled water containing 0.1% formic acid (solvent system A) and acetonitrile (CH_3_CN) containing 0.1% formic acid (solvent system B) in a gradient mode. The gradient program was used as follows: A/B (85:15, *v*/*v*) from 0 to 4 min, linear change from A/B (85:15, *v*/*v*) to A/B (0:100, *v*/*v*) from 4 to 5 min, A/B (0:100, *v*/*v*) from 5 min to 8 min, and linear change from A/B (0:100, *v*/*v*) to A/B (85:15, *v*/*v*) from 8 min to 8.1 min. The sample injection volume was 3 µL, flow rate was set to 0.3 mL/min, column temperature was set to 40 °C, and UV wavelength was set to 326 nm. The peak was identified based on retention time and comparison with the authentic reference compound injected.

### 2.8. Calibration Curve of Chlorogenic Acid

A series of calibration standard solutions of chlorogenic acid were prepared in methanol at concentrations of 6.25, 12.5, 25, 50, and 100 µg/mL. Each concentration was examined thrice. The calibration curve was produced by plotting the peak area under curve versus the concentration of the standard. The regression equation was used to determine the concentration of chlorogenic acid in samples.

### 2.9. Statistical Analysis

All experiments were conducted at least thrice. Based on the average value, the results are presented as the mean ± standard error of the mean. Data were evaluated using one-way analysis of variance with the SPSS statistics 29.0 (SPSS Inc., Chicago, IL, USA). A value of *p* < 0.05 indicated statistical significance.

## 3. Results

### 3.1. Effect of LH Water Extract on BV2 Microglial Cell Viability 

The MTT assay was performed to evaluate the maximal concentration of LH water extract without cytotoxicity in BV2 cells. When BV2 cells were cultured with 0.01, 0.05, 0.1, 0.3, and 0.5 mg/mL of LH for 24 h, no cytotoxicity was detected up to 0.3 mg/mL (Figure 1). Based on these results, subsequent experiments were conducted with 0.3 mg/mL as the maximum concentration, excluding 0.5 mg/mL.

### 3.2. Effect of LH Water Extract on LPS-Induced iNOS Expression and Nitrite Formation in BV2 Microglial Cells

Excessively released NO in the central nervous system (CNS) is associated with the development of neuroinflammation. Impaired activities of inflammatory factors such as iNOS and COX-2 exert a significant impact on the development of inflammatory diseases [16,17]. Real-time RT-PCR was conducted to assess whether LH water extract modulated LPS-induced mRNA levels of iNOS and COX-2 in BV2 cells. In the cell group pre-treated with LH water extract, the expression of iNOS was found to be gradually downregulated in a dose-dependent manner, compared to the cell group stimulated with LPS alone. In particular, it decreased significantly at 0.05, 0.1, and 0.3 mg/mL (Figure 2A). The group treated with LPS alone had higher mRNA levels of COX-2 than the control group, and LH water extract failed to reduce the expression of COX-2 activated by LPS (Figure 2B). Because microglia express iNOS and promote the NO release in response to inflammatory stimuli [17], the effect of LH water extract on regulating excessive NO secretion was investigated. LH extract significantly suppressed NO secreted by LPS at all concentrations (Figure 2C).

### 3.3. Effect of LH Water Extract on LPS-Induced Pro-Inflammatory Cytokine Release in BV2 Microglial Cells

Because activated microglia promote the secretion of pro-inflammatory cytokines contributing to neurodegenerative mechanisms [18], we investigated whether LH water extract modulated LPS-induced pro-inflammatory cytokine release in BV2 cells. In the LPS-activated cell group, the mRNA levels of IL-1β, IL-6, and TNF-α were markedly upregulated, and LH water extract predominantly diminished the IL-1β and TNF-α release. In contrast, LH water extract did not affect the secretion of IL-6 (Figure 3).

### 3.4. Effect of LH Water Extract on LPS-Induced MAPK and NF-κB Activation in BV2 Microglial Cells

MAPK signaling is responsible for the production of pro-inflammatory cytokines and contributes to regulating the processes of inflammation [19]. In cell groups stimulated with LPS alone for 15, 30, and 60 min, increased phosphorylation of ERK1/2, JNK, and P38 and degraded IκBα were observed. As shown in Figure 4, phosphorylation of JNK was prevented in the group pretreated with 0.3 mg/mL of LH water extract. However, it did not interfere with LPS-induced ERK1/2 and P38 phosphorylation, nor did it affect IκBα degradation.

### 3.5. Identification of Chlorogenic Acid in LH 

Next, we decided to identify the component contributing to the anti-neuroinflammatory activity of LH extract using the UPLC assay. As shown in Figure 5, UPLC analysis of LH extract revealed a major peak (RT 1.508 min), which was identified as chlorogenic acid. The concentration of chlorogenic acid was calculated by using calibration curve (R^2^ = 0.999986, y = 29.649x + 618). It was ascertained that LH extract contained approximately 57.84 ± 0.09 mg/g of chlorogenic acid.

## 4. Discussion

Although each neurodegenerative disorder has a different pathological mechanism, all of them share common features such as persistent neuroinflammation [20]. Neuroinflammation indicates an inflammatory response in the CNS triggered by several pathological insults and is primarily managed by innate immune cells [21]. Microglia, which are resident innate immune cells in the CNS, play a pivotal and active role in the pathophysiology of brain diseases by regulating immune responses [22]. In the resting state, microglia contribute to immune surveillance and maintenance of neuronal homeostasis. However, when stimulated by pathological insults, such as LPS, TNF-α, and interferon-γ (IFN-γ), they lose their homeostatic function and release substances involved in neuroinflammation [21,23]. In addition, microglia and immune signaling not only have limited roles such as in controlling the secreting inflammatory molecules in disease development but also modulate protein aggregation and synaptic loss in the early stages of neurodegeneration [24,25,26]. Therefore, neuroinflammation and microglia are attracting wide attention in the field of neurodegenerative disorder treatment.

The NO/NOS system is linked to the pathophysiology of numerous diseases, including neurodegenerative disorders, and is cardinal for the nervous system [27]. NO can be synthesized by several members of the nitric oxide synthases (NOS) family, which are classified into two groups based on their dependence on Ca^2+^-calmodulin levels [28]. iNOS is highly expressed due to the activated state of microglia and other cells by inflammatory stimuli, whereas the expression of neuronal NOS (nNOS) and endothelial NOS (eNOS) increases depending on the intracellular Ca^2+^ concentration [29,30]. NO synthesized by iNOS and secreted from activated microglia is immoderately generated during the neuroinflammatory process; excessive NO inhibits neuronal respiration, resulting in excitotoxic neuronal cell death through glutamate release [31,32]. Therefore, regulation of iNOS and NO is considered crucial in neuroinflammation. In the present study, we confirmed that the LH extract regulates neuroinflammation by impeding the formation of iNOS and nitrite in LPS-activated microglial (Figure 1 and Figure 2).

Cytokines, which constitute a significant component of immune and neuromodulatory milieu secreted by activated microglia, protect the CNS by managing innate defense mechanisms [33]. However, failure to protect against the production of excessive cytokines derived from activated microglia can lead to neuronal death and neurodegeneration following apoptosis, excitotoxicity, immune activation, and cytotoxicity [18]. As shown in Figure 3, LH water extract exerted an anti-neuroinflammatory effect by reducing LPS-induced IL-1β and TNF-α release in BV2 cells. IL-1β is maintained at a low level under physiological brain conditions; however, activated microglia markedly release IL-1β in the CNS under neurodegenerative conditions [34]. Moreover, overexpressed IL-1 in Alzheimer’s brain is associated with protein aggregation [35]. Because IL-1β contributes to the severity of neurodegenerative disorders, studies are being conducted to modulate IL-1β or IL-1β signaling in the treatment of neuroinflammation [36]. TNF-α is a representative cytokine that regulates CNS inflammatory response by participating in each stage of excitotoxic cell death and neuroinflammation under pathological conditions as well as inflammation in the human body [37]. Accordingly, synthesized TNF-α inhibitors have been suggested as treatments for neurodegenerative disorders; however, their application is difficult due to the associated toxicity and adverse effects [38]. In this regard, LH, a plant extract, is a potential therapeutic for improving neurodegenerative disorders by interrupting IL-1β and TNF-α release.

MAPK-and NF-κB-activated signaling pathways are typical mechanisms producing inflammatory molecules during LPS-induced inflammation [39]. In particular, inhibitors of MAPK activation are regarded as potent anti-inflammatory agents due to their remarkable efficacy in blocking cytokine signaling and inhibiting the synthesis of inflammatory mediators [40]. Investigation of the pathway by which the anti-neuroinflammatory action of LH water extract is executed revealed that LH water extract inhibited the LPS-induced JNK pathway in BV2 cells (Figure 4). JNK, one of the MAPK superfamily members, activates the transcription of inflammation-related genes, including *TNF-α*, when activated by inflammatory signals in glial cells [41,42]. Accordingly, JNK is a signal transduction pathway intricately related to neurodegenerative disorders and is essential for processes associated with CNS inflammation [41,43]. Because JNK inhibitors have neuroprotective properties [43], LH is predicted to alleviate neuroinflammation and exert neuroprotection by preventing JNK phosphorylation.

Chlorogenic acid is a representative bioactive component of LH [44]. UPLC analysis revealed that the LH used in this study also contained chlorogenic acid (Figure 5). Additionally, chlorogenic acid was confirmed to have anti-inhibitory effects similar to those of LH water extract in BV2 activated with LPS (Appendix A). Chlorogenic acid is known to regulate the M1/M2 polarization level of microglia, to inhibit microglial activation, and to exert immunomodulatory activity in an LPS-induced neuroinflammation model [45]. In addition, it demonstrated excellent anti-inflammatory efficacy by suppressing LPS-induced nitrite, iNOS, and pro-inflammatory cytokine expression in macrophages [11]. Based on these reports, it is assumed that chlorogenic acid contributes, to a certain extent, to the effectiveness of LH in improving neuroinflammation.

Currently, drug-based methods in the treatment of neurodegenerative disorders are associated with diverse side effects such as insomnia and hormonal imbalance. Thus, safe and effective herb-based medicines are receiving considerable attention [46,47]. FH is a natural material that has been used as a raw material for medicines or tea since ancient times, but LH has been relatively neglected. This study is the first to confirm the possibility of improving neuroinflammation of LH. According to a recent study, the importance of LH has been underestimated despite its chemical composition being very similar to FH, and even despite the production of LH being higher than that of FH. Thus, utilizing LH as a raw material could generate economic benefits [10,48]. Therefore, LH might be useful not only from a pharmacology perspective but also from the viewpoint of economics and natural utilization.

## 5. Conclusions

Taken together, LH water extract exhibited anti-neuroinflammatory effects by reducing the expression of iNOS, NO, IL-1β, and TNF-α through the inhibition of JNK phosphorylation in LPS-stimulated BV2 microglial cells. The results suggest that LH has a high therapeutic potential or could be used as a raw material for improving neuroinflammation. However, follow-up research is needed.

## Figures and Tables

**Figure 1 nutrients-16-03954-f001:**
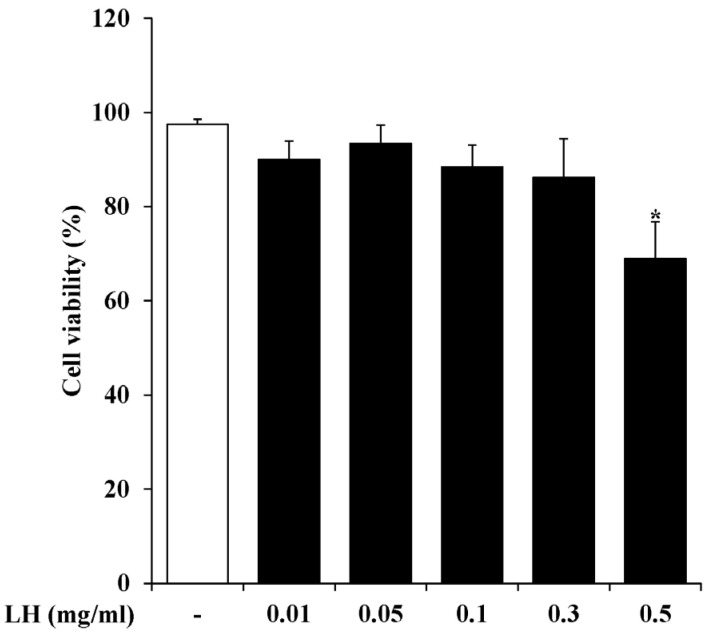
The cytotoxicity of the leaf of honeysuckle (LH) water extract in BV2 microglial cells. Cells were cultured with LH water extract at indicated concentrations, and after 24 h, cell viability was assessed using the 3-(4,5-dimethylthiazol-2-yl)-2,5 diphenyl tetrazolium bromide (MTT) reagent. Data are expressed as means ± standard error of the mean (SEM). Results are representative of three experiments. * *p* < 0.05 vs. saline treatment alone.

**Figure 2 nutrients-16-03954-f002:**
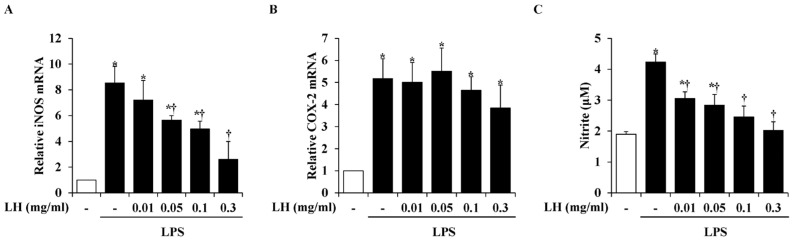
Effect of leaf of honeysuckle (LH) water extract on the production of inflammatory mediators in BV2 microglial cells activated by lipopolysaccharide (LPS). BV2 cells were pre-treated with LH water extract at indicated concentrations for 1 h and next co-incubated with LPS (1 μg/mL) for 6 h. Afterward, mRNA levels of (**A**) inducible nitric oxide synthase (iNOS) and (**B**) cyclooxygenase (COX)-2 were assessed using real-time reverse transcription-polymerase chain reaction (RT-PCR). (**C**) The cells were pre-treated with LH water extract and co-incubated with LPS (1 μg/mL) for 24 h. Next, nitrite production was evaluated by the Griess reaction. Data are expressed as means ± standard error of the mean (SEM). Results are representative of three experiments. * *p* < 0.05 vs. saline treatment alone; † *p* < 0.05 vs. LPS treatment alone.

**Figure 3 nutrients-16-03954-f003:**
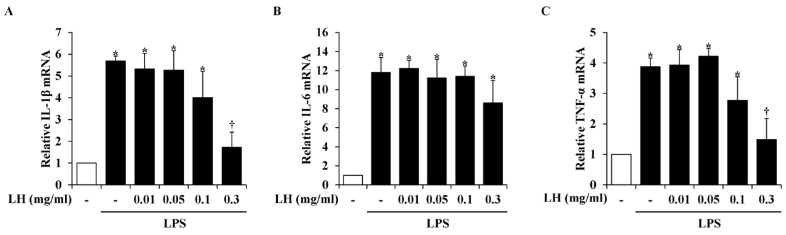
Effect of leaf of honeysuckle (LH) water extract on the secretion of inflammatory cytokines in BV2 microglial cells. The cells were pre-treated with LH water extract at indicated concentrations for 1 h and next co-incubated with lipopolysaccharide (LPS) (1 μg/mL) for 6 h. The mRNA levels of (**A**) interleukin (IL)-1β, (**B**) IL-6, and (**C**) tumor necrosis factor (TNF)-α were assessed using real-time reverse transcription-polymerase chain reaction (RT-PCR). Data are expressed as means ± standard error of the mean (SEM). Results are representative of three experiments. * *p* < 0.05 vs. saline treatment alone; † *p* < 0.05 vs. LPS treatment alone.

**Figure 4 nutrients-16-03954-f004:**
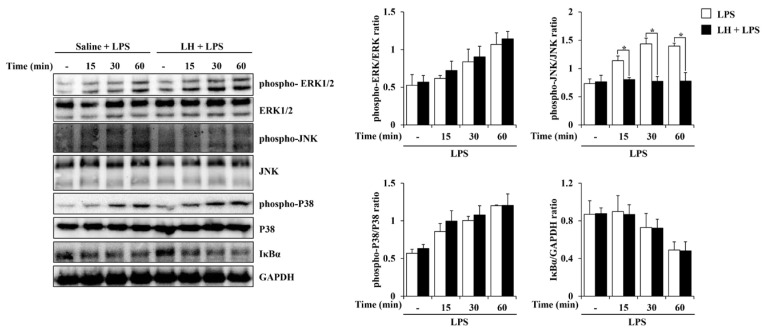
Effect of leaf of honeysuckle (LH) water extract on lipopolysaccharide (LPS)-induced mitogen-activated protein kinase (MAPK) phosphorylation and inhibitory κBα (IκBα) degradation in BV2 microglial cells. The cells were pre-treated with 0.3 mg/mL LH or saline for 1 h and next co-incubated with LPS (1 μg/mL) for 15, 30, and 60 min. The samples were derived from the same experiment. Data are expressed as means ± standard error of the mean (SEM). Results are representative of three experiments. * *p* < 0.05 vs. LPS treatment alone.

**Figure 5 nutrients-16-03954-f005:**
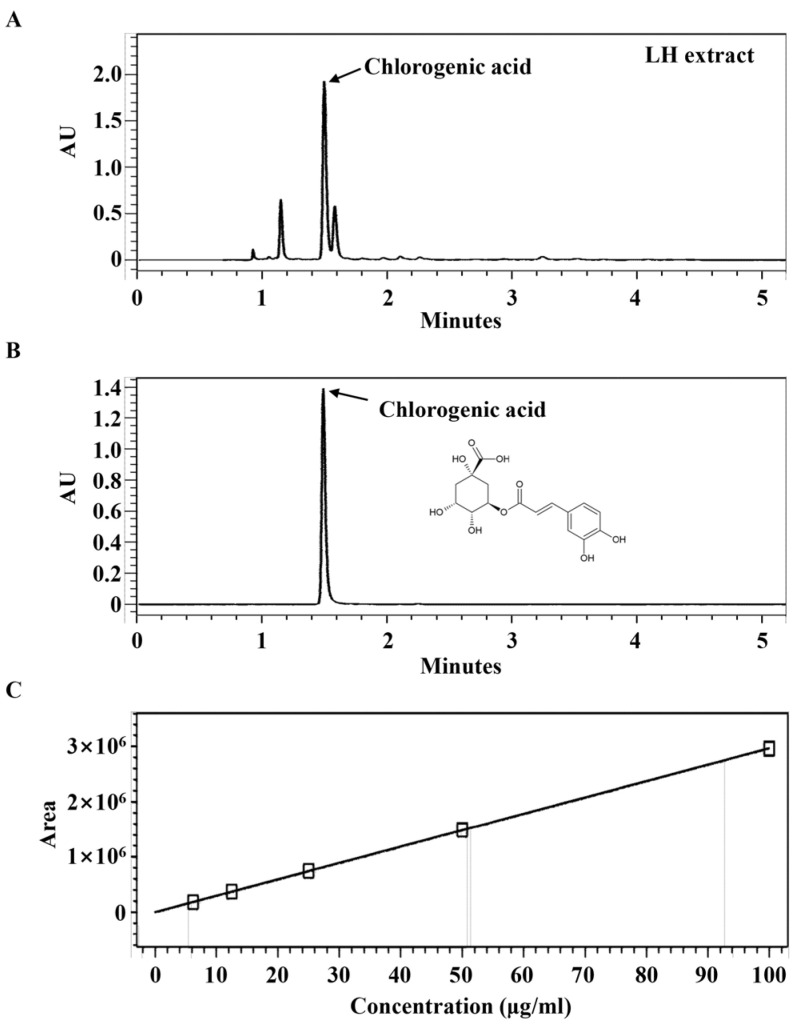
Ultra-performance liquid chromatography (UPLC) analysis of leaf of honeysuckle (LH) extract. Chromatogram of (**A**) LH extract and (**B**) chlorogenic acid as standards at 326 nm. (**C**) Calibration curve of chlorogenic acid.

## Data Availability

The datasets used and/or analyzed during the current study are available from the corresponding author on reasonable request. The data are not publicly available due to the data are part of an ongoing study.

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
