# Peer review of "Anti-Inflammatory Effects of Honeysuckle Leaf Against Lipopolysaccharide-Induced Neuroinflammation on BV2 Microglia"

_nutrients, 2024, doi:10.3390/nu16223954_

Round 1

Reviewer 1 Report (Previous Reviewer 2)

Comments and Suggestions for Authors

The authors did not respond to the comment related to the novelty and contribution to the field.

Comments on the Quality of English Language

Although the authors provided a certificate of English editing, the text has not improved significantly. In some cases, the original phrasing was clearer. While many changes appear to have been made (as indicated by the red text), these revisions are minor, and the manuscript still needs further English improvement.

Author Response

Reviewer 2 Report (Previous Reviewer 1)

Comments and Suggestions for Authors

The study aimed to investigate the neuroprotective activity of a water extract from the leaves of Lonicera japonica in microglial cells stimulated by lipopolysaccharide as an inflammatory agent. The subject is worthy of investigation, as the plant has been used in traditional medicine for ages.

I reviewed the manuscript previously, and the authors have addressed all the concerns pointed out. I have only two minor suggestions:

1)      Section 2.7: For identification, the UV-Vis spectrum of the standard and extract components should be compared.

2)      Remove Figure 5c. The information regarding calibration curve can be included to the text

Author Response

Reviewer 3 Report (New Reviewer)

Comments and Suggestions for Authors

Review on the manuscript of Kweon B et al., (nutrients-3222368): “Anti-inflammatory effects of honeysuckle leaf against lipopolysaccharide-induced neuroinflammation on BV2 microglia”.

In this study, the Authors investigated the anti-inflammatory effects of a water extract from leaf honeysuckle (LH) in LPS-stimulated BV2 cells. They observed that the LH extract, containing 57.84 ± 0.09 mg/g of chlorogenic acid, attenuated LPS-induced iNOS, IL-1β, and TNF-α mRNA expression levels, as well as nitrite levels, at non-cytotoxic concentrations. Additionally, the LH extract also reduced LPS-induced phosphorylation of JNK. The Authors concluded that the LH water extract could potentially improve neuroinflammation.

Overall, I find this topic to be of great interest, as neuroinflammation plays a critical role in the progression of neurodegenerative diseases. Therefore, targeting this process presents a promising therapeutic strategy for these conditions. I believe the Authors have addressed the main question posed. The manuscript is well-written and well-organized, demonstrating rigor in presenting the results. The issues I have identified with the current version of the manuscript are listed below. I hope the Authors find the following comments and suggestions helpful.

1 - For the experiments, the Authors always used the same extract. Did the Authors characterize the extract to identify which compounds were present and their concentration (only chlorogenic acid was identified)? Without such information, it would be difficult to replicate similar experiments in the future.

2 - The Authors saw that LPS-induced IL-1β mRNA expression was efficiently attenuated by the LH extract. However, no effect of the LH extract on LPS-induced IL-6 mRNA expression was observed. Could the Authors provide an explanation for this distinct effect?

3 - In Figure 4, the WB image for ERK1/2 is not suitable for inclusion in the figure. I recommend that the Authors replace it with a different image.

4- The Authors performed UPLC analysis to identify the putative compound of the LH extract contributing to its anti-neuroinflammatory activity and found that chlorogenic acid was the main component. However, it cannot be concluded that this compound is solely responsible for the anti-inflammatory effects of the LH extract. I recommend that the Authors repeat the experiments using pure chlorogenic acid. Without such data, this conclusion cannot be drawn. In addition, why were the other components in the LH extract not identified?

Round 2

Reviewer 3 Report (New Reviewer)

Comments and Suggestions for Authors

Second review on the manuscript of Kweon B et al., (nutrients-3222368): “Anti-inflammatory effects of honeysuckle leaf against lipopolysaccharide-induced neuroinflammation on BV2 microglia”.

In this study, the Authors investigated the anti-inflammatory effects of a water extract from leaf honeysuckle (LH) in LPS-stimulated BV2 cells. They observed that the LH extract, containing 57.84 ± 0.09 mg/g of chlorogenic acid, attenuated LPS-induced iNOS, IL-1β, and TNF-α mRNA expression levels, as well as nitrite levels, at non-cytotoxic concentrations. Additionally, the LH extract also reduced LPS-induced phosphorylation of JNK. The Authors concluded that the LH water extract could potentially improve neuroinflammation.

This represents a second version of the manuscript after peer review. After carefully reading the revised manuscript and the authors’ response letter, I still believe that some points need to be clarified for the manuscript. The issues I identified in the current version of the manuscript are listed below. I hope the authors find the following comments and suggestions helpful

1 - I understand that chlorogenic acid is the main component in the extract. Based on the literature, we can assume that chlorogenic acid plays a role in the protective effects observed. However, I believe that a full characterization of the extract or experiments with purified chlorogenic acid are essential to draw clear conclusions.

Author Response

This manuscript is a resubmission of an earlier submission. The following is a list of the peer review reports and author responses from that submission.

Round 1

Reviewer 1 Report

Comments and Suggestions for Authors

In the manuscript, the authors investigated the neuroprotective effects of a water extract from the leaf of Lonicera japonica in microglial cells induced by lipopolysaccharide. The study involved the impact on mediators of inflammation and the mitogen-activated protein kinase pathway. The study is interesting and provided new data regarding L. japonica. However, the manuscript needs some improvement.

Detailed comments:

1)    The structure of the Abstract should be improved. The background (lines 16-23) is too long. Add more information regarding the results. Include information about the determination of chlorogenic acid. Line 26: do not start the sentence with "and".

2)    Introduction:

- Could you explain the activity you mentioned: „ (….) disperses wind-heat. For this reason, they have been mainly used to treat wind-dampness pain….” – what kind of action did you mean? Antipyretic?

- „LSH contain diverse components including phenolic acids…” –  what other components were found in the plant? Add data regarding the phytochemical composition of the plant.

- Line 69: unnecessary „that”

- Add information on why the authors focused the phytochemical investigation on chlorogenic acid.

3) Materials and Methods: “20 times the amount of distilled water (600 ml) was added…” -  unclear. 20 x 600 mL?

2.7. section: Line 155: “to dissolve the extracts and standards” – use singular form. Only one extract and standard were used. Add information on how the identification of chlorogenic acid was done. Include details regarding the quantitative analysis.

4) Results: “The MTT assay was performed to explore the maximum biologically efficient concentration of non-toxic LH water extract in BV2 cells.” – unprecise statement. This stage aimed to assess the maximal concentration which was not cytotoxic.

“3.5. Analysis components isolated from LH extracts” – unprecise statement. Chlorogenic acid was not isolated from the extract. On what basis did the authors confirm the identity of the compound? Was the purity of the peak checked?

5) Figures: add the explanation for all abbreviations in figure legend. Figures should be self-explanatory. Fonts on figures should be larger – they are hardly visible. Figure 5c is unnecessary. Calibration data should be added to the text (to Materials and Methods). Figure 5 legend: Add information on the wavelength at which the chromatogram was recorded.

There are a few editorial errors:

-          Lack of capital letters for ml (should be; mL)

-          Line 160: “(CH3CN)”

-          Correct the references according to the journal’s requirements.

Reviewer 2 Report

Comments and Suggestions for Authors

The authors investigated the anti-inflammatory effect of the extract obtained from the Honeysuckle leaves (Lonicera japonica Thunb.). The authors have shown the extract reduces the LPS-induced increase in iNOS gene expression and NO production, and TNF-alpha and IL-1beta gene expression likely by modulating JNK signalling.

Although the study is performed adequately, previous research has already shown that Lonicera japonica Thunb. holds considerable promise as an anti-inflammatory and neuroprotective agent due to its ability to modulate key inflammatory pathways, making it a potential candidate for the treatment of neuroinflammatory and neurodegenerative diseases. Similar findings have also been reported for chlorogenic acid, one of the main components of the extract. In my opinion, this study does not add much novelty to the field.

Major comments:

1. English language reqires detailed revision

2. All JNK blots performed should be provided, on the representative blot provided it is hard to see changes that are indicated on graphical representation

Minor comments:

Title -  study was not performed on neuronal cells, neuroprotective effect is not appropriate term

L47-neuroimmune and neuroinflammatory processes - the same terminology

Ref. 8?

L95 - the MTT reagent was treated in - English

L96 - for 30 min – MTT is usually kept on cells for 3-4 hours

L107 - based on standard diluted sodium nitrite (NaNO2) in stages – unclear

References are not cited uniformly

L301 – ref. 42 should be related to neurodegenerative conditions

Comments on the Quality of English Language

Needs to be revised carefully